# Sequencing and Phylogenetic Analysis of Chloroplast Genes in Freshwater Raphidophytes

**DOI:** 10.3390/genes10030245

**Published:** 2019-03-22

**Authors:** Ingrid Sassenhagen, Karin Rengefors

**Affiliations:** 1Laboratoire d’Océanologie et des Geosciences, UMR CNRS 8187, Université du Littoral Côte d’Opale, 62930 Wimereux, France; 2Aquatic Ecology, Department of Biology, Lund University, 22362 Lund, Sweden; karin.rengefors@biol.lu.se

**Keywords:** raphidophytes, chloroplast genome, whole-genome-amplification, phylogeny, LIPOR

## Abstract

The complex evolution of chloroplasts in microalgae has resulted in highly diverse pigment profiles. Freshwater raphidophytes, for example, display a very different pigment composition to marine raphidophytes. To investigate potential differences in the evolutionary origin of chloroplasts in these two groups of raphidophytes, the plastid genomes of the freshwater species *Gonyostomum semen* and *Vacuolaria virescens* were sequenced. To exclusively sequence the organelle genomes, chloroplasts were manually isolated and amplified using single-cell whole-genome-amplification. Assembled and annotated chloroplast genes of the two species were phylogenetically compared to the marine raphidophyte *Heterosigma akashiwo* and other evolutionarily more diverse microalgae. These phylogenetic comparisons confirmed the high relatedness of all investigated raphidophyte species despite their large differences in pigment composition. Notable differences regarding the presence of light-independent protochlorophyllide oxidoreductase (LIPOR) genes among raphidophyte algae were also revealed in this study. The whole-genome amplification approach proved to be useful for isolation of chloroplast DNA from nuclear DNA. Although only approximately 50% of the genomes were covered, this was sufficient for a multiple gene phylogeny representing large parts of the chloroplast genes.

## 1. Introduction

Plastids are likely derived from a single endosymbiotic event incorporating a free-living cyanobacterium into the ancestor of the *Archaeplastida* (glaucophytes, red algae and green algae) [1]. The chloroplasts of both red algae and green algae were subsequently transferred to other lineages by secondary endosymbiosis [2]. Red algae appear to have been taken up as endosymbionts only once, giving rise to a diverse group called chromalveolates. Additional layers of complexity come from recurring plastid loss and replacement. Some photosynthetic lineages have thus evolved from ancestors with a plastid of different origin, meaning that an ancestral plastid has been replaced with a new one [2]. Such replacement has taken place in several dinoflagellates by tertiary endosymbiosis with other chromalveolates, or serial secondary endosymbiosis with a green alga. This study focuses on plastids that carry out photosynthesis, and they will be referred to as chloroplasts throughout the text independent of their evolutionary acquisition.

Raphidophytes are eukaryotic, unicellular algae belonging to the chromalveolates. In particular, the marine species *Heterosigma akashiwo* and *Chattonella subsalsa* have received much scientific attention as they form harmful algal blooms and can cause extensive fish kills [3]. However, the freshwater species, such as *Gonyostomum semen* and *Vacuolaria virescens*, have been often overlooked and are less studied due to their lack of toxins. Most phylogenetic and evolutionary studies of this class are, therefore, limited to the marine species.

A conspicuous difference between freshwater and marine raphidophytes is their large difference in coloration, marine taxa being brownish when observed under the microscope while freshwater taxa appear bright green. Analyses of pigment composition of *G. semen* and *V. virescens* revealed the presence of unusual pigments and identified major differences compared to marine raphidohytes [4]. The marine genera *Chattonella, Heterosigma* and *Fibrocapsa* share the pigments chlorophyll *a*, chlorophyll *c1c2*, violaxanthin, zeaxanthin, *a* and *b* carotene with *G. semen* and *V. virescens*, but also contain auroxanthin and fucoxanthin [5], which probably generates the typical brown color of these taxa. The freshwater raphidophytes *G. semen* and *V. virescens* additionally possess trans-neoxanthin, cis-neoxanthin, diadinoxanthin and alloxanthin [4]. The origin of these different pigments in raphidophytes, especially the presence of alloxathin that typically occurs in cryptophytes, is currently unknown and might result from tertiary endosymbiosis.

To illuminate the evolutionary origin of freshwater raphidophyte chloroplasts, we selectively sequenced the respective genomes of *G. semen* and *V. virescens* after isolation of the organelles and whole-genome-amplification, thereby avoiding sequencing of the large nuclear genome. Assembled and annotated chloroplast genes were phylogenetically compared to genes of the marine raphidophyte *H. akashiwo* and other evolutionarily diverse microalgae.

## 2. Materials and Methods 

The chloroplast genomes were sequenced from the *G. semen* culture GSMJ21 (Lund University, Sweden) and the *V. virescens* culture SAG 1195-1 (Culture Collection of Algae, Göttingen, Germany). For isolation of the chloroplasts, 1.5 mL of algal culture was pelleted by centrifugation and, after removal of the supernatant, resuspended in 1 mL breaking buffer (300 mM sorbitol, 50 mM Hepes-KOH (pH 7.5), 2 mM Na-EDTA, 1 µL MgCl_2_, 1% bovine serum albumin (BSA)) [6]. The algal cells were disrupted by vortexing with glass beads for 5 min. 10–20 chloroplasts were isolated from each sample by micropipetting under an inverted microscope (Nikon Eclipse TS100; Melville, NY, USA) at 400× magnification into polymerase chain reaction (PCR) tubes with 3 µL phosphate buffered saline (PBS). The whole chloroplast genomes from the pooled 10–20 organelles were amplified by multiple displacement amplification (MDA) using the Qiagen RELPI-g kit (Qiagen, Hilden, Germany). Sequencing libraries of two samples from each species were constructed and paired-end sequenced on a partial Illumina MiSeq lane at the Department of Biology DNA sequencing facility in Lund (Sweden).

The quality of the Illumina reads was checked using the software FastQC [7]. Low quality sequences were trimmed or removed with the program Trimmomatic-0.36 [8] using leading and trailing cut-off values of 15 bp and a sliding-window approach with a window size of 10 bp and a quality threshold of 20. Reads from samples of the same species were pooled for assembly with IDBA-ud [9], SPAdes [10] and metaSPAdes [11] and the resulting assemblies were assessed with quast 4.5 [12]. IDBA-ud settings included pre-correction of reads before assembly, a minimum k value of 30 and a maximum k value of 70, k-mer increments of 10 bp, and a seed kmer size of 30 bp. SPAdes and meta-SPAdes were run with default settings and k-mer sizes of 21, 33, 55, 77 and 99 bp. Assemblies were also repeated for reduced datasets (10%, 20%, 25% and 30% of reads) due to very high (>1000×) and uneven coverage of chloroplast contigs.

After each assembly, the unconnected contigs were organized into biologically relevant units (bins) with Anvio (v2.2.2) [13] using information about GC content (approx. 30% GC expected based on *H. akashiwo* chloroplast genomes), coverage and sequence similarity. Thereby, contigs belonging to the chloroplasts were separated from potential nuclear DNA and bacterial contaminations. All contigs in bins that potentially represented the chloroplast genomes were compared with the NCBI nucleotide database using blastn [14]. To improve the assemblies, the coverage of the chloroplast contigs was homogenized by extracting the corresponding reads with the program Bowtie2 [15], and python and perl scripts from Albertsen et al. [16]. The size of these contig-specific read sets was reduced to approximately 100× coverage by splitting them according to their original read depth. The remaining reads were re-assembled with SPAdes or metaSPAdes.

To compare the contribution from replicated libraries of the same species to the chloroplast genome assemblies, trimmed raw reads were mapped with the software Bowtie2-2.3.4.3 [17] against the assembled contigs using default settings. These alignments were analysed with samtools-1.9 [18]. 

Phylosift [19] was used to identify marker genes in the chloroplast contigs from *G. semen* and *V. virescens*. The chloroplast contigs were afterwards translated into amino acid sequences using the program prodigal [20]. These sequences were functionally annotated in KAAS (KEGG Automatic Annotation Server) [21] using the single-directional best hit method. To identify potentially unassembled fragments of light-independent protochlorophyllide oxidoreductase (LIPOR) genes (*chlB, chlN, chlL*) in the raw reads from *G. semen*, these reads were mapped with Bowtie2 against the chloroplast contigs of *V. virescens* that contained the genes of interest. Annotated amino acid sequences were aligned with MUSCLE [22] using default settings in the program Geneious 11.1.5 [23] and trimmed using the software trimAl applying a gap threshold of 0.75 [24].

Maximum likelihood trees were build for marker genes from Phylosift, individual and 53 concatenated amino acid sequences annotated with KAAS (*16S rRNA, ef1aLike, hsp70, rplA, rplV, rplB, rpsI, rplC, rpsS, rpsG, rplP, rp L25/L23, rplF, rplK, rplE, rp S12/S23, rpsK, rpsH, rp L18P/L5E, rplM, rp L24, psaA, psaB, psaC, psaD, psaE, psaF, psaL, psbA, psbB, psbC, psbD, psbE, psbH, psbL, psbV, rbcS, rbcL, rpoA, rpoB, rpoC1, rpoC2, ilvB, sufB, petJ, chlB, chlN, chlL, atpA, atpD, atpE, clpC, groEL*) using RAxML [25]. The RAxML settings included rapid bootstrap analysis and automatic determination of protein substitution model while the number of distinct starting trees was based on bootstrapping criteria. For these trees, corresponding sequences from 47 reference organisms, selected to cover the genetic diversity within most microalgae lineages, were downloaded from NCBI. All trees were visualized with the online application iTOL [26]. tRNAs in the chloroplast bins were identified using the online program tRNAscan-SE [27]. Similarity and approximate coverage of the *H. akashiwo* chloroplast genome by *G. semen* and *V. virescens* contigs was assessed using blastn and visualized with the online application CGview [28]. 

## 3. Results

### 3.1. Processing of Reads

For *G. semen*, the two libraries yielded 702,765 and 706,755 forward and reverse reads (in total 1,409,520 reads), while 392,682 and 397,490 forward and reverse reads were generated for *V. virescens* (in total 790,172 reads) (Table 1). Different assembly approaches yielded the best results for *G. semen* and *V. virescens* probably due to differences in read quality and coverage. The reads from the *V. virescens* sample were assembled with metaSPAdes to 7172 contigs with on average 75× coverage. The best assembly was achieved for *G. semen* by assembling 25% of the reads with SPAdes, identification of chloroplast contigs through binning, homogenization of coverage of these selected contigs and reassembly of the remaining reads with SPAdes to 1649 contigs. After binning with Anvio and identity control with blastn, 35 contigs with a total length of 88,279 bp were identified as belonging to the chloroplast genome of *V. virescens* and 39 contigs with a total length of 74,603 bp to *G. semen* (Table 1). In comparison, the size of the chloroplast genome of the marine raphidophyte *H. akashiwo* ranges between 159,321 and 160,152 bp [29,30] suggesting approximately 50% completeness of the novel freshwater raphidophyte chloroplast genomes. The chloroplast contigs from *G. semen* and *V. virescens* had a mean coverage of 613× and 472× respectively with individual contigs having more than 1000× coverage. The majority of the contigs had in both cases a length between 1000–5000 bp and only 3 and 1 contigs respectively in each assembly were longer than 5000 bp. The assembled contigs can be found at NCBI under the accession numbers MK658836 and MK674494.

No contigs from the nuclear genome of the two algal species were identified in the other bins, which instead consisted of contigs from various bacterial species. Based on mapping with Bowtie2, the majority of reads, 81% for *G. semen* and 85% for *V. virescens*, belonged to bacteria despite the high coverage of chloroplast contigs. Reads from replicated libraries were similarly distributed over the individual chloroplast contigs, meaning that contigs with high coverage attracted many reads from both libraries, while relatively few reads from both replicates mapped to contigs with lower coverage.

### 3.2. Annotation

Phylosift identified 23 marker genes in the contigs from *G. semen* including the 16S rRNA, the RNA polymerase β’ subunit (V136), the translation elongation factor 1, heat shock protein 70, cytochrome b6 (petB) and 18 ribosomal subunits. In the contigs from *V. virescens*, the 16S rRNA, the translation elongation factor 1, heat shock protein 70 and 26 ribosomal subunits were found. tRNAscan-SE found 10 tRNA sequences in the *G. semen* contigs (Ser, Gly, Phe, Ile, 2× Ala, 2× Leu, Asn, Asp) and seven in the *V. virescens* contigs (His, Arg, Ile, Ala, Val, Arg, Gly).

Prodigal identified 95 potential protein-coding sequences for *G. semen* and 115 for *V. virescens*. Out of these, 72 amino-acid sequences could be annotated with KAAS for *G. semen* and 84 for *V. virescens*. All genes were previously described in other microalgal chloroplast genomes and most were also found in the closest sequenced relative *H. akashiwo.* The complete chloroplast genome of *H. akashiwo* contains 156 protein coding genes, 130 with assigned function. Although the chloroplast genomes from this study remained incomplete preventing synteny analyses with reference genomes, the order of genes is likely very similar to *H. akashiwo,* as the freshwater raphidophyte plastid contigs mapped very well to large parts of the chloroplast genome of the marine raphidophyte (Figure 1). High sequence similarity was observed, especially in coding genomic regions with higher than average GC content (30.5%).

### 3.3. Phylogenetic Analyses

The phylogenetic analyses of 53 different chloroplast protein sequences using maximum likelihood trees showed a high relatedness of *G. semen*, *V. virescens* and *H. akashiwo* (Figure 2). Individual and concatenated sequences from the raphidophyte species clustered together in all phylogenetic analyses. A maximum likelihood tree based on 53 chloroplast genes concatenated and trimmed to 14,896 amino acids confirmed the monophyly of *stramenopiles*. The raphidophytes represented the sister group to *xantophyceae* and *phaeophyceae*. The arrangement of genes in the different chloroplast genomes could not be compared due to the incompleteness of the *G. semen* and *V. virescens* genomes and the limited length of the assembled contigs.

One significant difference in gene composition compared to the completely sequenced chloroplast genomes of *H. akashiwo* [29,30] was, however, identified. The genes for the three subunits of the light-independent protochlorophyllide oxidoreductase (LIPOR: *chlB, chlL, chlN*) are encoded in the chloroplast genome of *V. virescens*, but were not identified in *G. semen*. The genes have previously been found in the marine raphidophyte *Chattonella subsala*, but appear to be absent in *H. akashiwo* and all diatoms. However, the genes are present in other *stramenopiles* such as phaeophyceae and xanthophyceae (Figure 2), suggesting independent loss of these genes in different taxonomic lineages [31].

## 4. Discussion

The whole-genome-amplification approach chosen for this study enabled de-novo sequencing of large parts of the chloroplast genome of two freshwater raphidophyte species. The advantage of this method is that it allows sequencing of the chloroplasts without wasting reads on the large nuclear genome (likely >100 pg per cell, [32]) thereby attaining very high sequencing depth of the targeted genomes. Since the chloroplasts from each library were acquired from single cells, we demonstrate that the method can be useful for taxa that cannot be cultured. Overall, the sequencing data yielded approximately 50% of the chloroplast genomes and 100 protein-coding sequences per species, showing that the approach allows for downstream phylogenomic analyses. 

However, the assembled contigs in this study were on average very short, which is likely caused by limitations of multiple displacement amplification (MDA) of the chloroplast genomes. Previous studies have shown that MDA is very sensitive to reaction gain, meaning that overrepresented fragments get amplified more throughout the reaction [33,34]. Thus, high gain potentially led to a strong bias towards certain genomic regions in our study and limited amplification of other parts of the genome. Similarly biased amplification was observed in both replicates of the sequenced species suggesting a recurring bias towards certain genomic regions such as areas with high GC content. This suggests that additional replicates would not necessarily have improved the chloroplast genome recovery. However, decreased reaction volumes achieved through nanoliter reactors [35] or emulsion-based amplification [36] might result in more uniform and accurate MDA of single cells. Furthermore, new single-cell whole genome amplification approaches, such as the MALBAC [37] or WGA-X [38], promise less bias and higher uniformity. 

A major concern when using whole-genome-amplification methods is amplification of non-targeted DNA, such as bacterial and human DNA. Binning of contigs based on similarity, GC-content and coverage allowed distinguishing between chloroplast and bacterial contigs. Treatment of the algal cultures with antibiotics to decrease the abundance of bacteria in the medium prior to the isolation of chloroplasts could likely further improve this approach.

Despite not recovering complete genomes, assembly of many chloroplast genes allowed phylogenetic comparisons with marine raphidophytes and other microalgae. The phylogenetic trees of individual and concatenated chloroplast genes confirmed the high relatedness of freshwater and marine raphiodphyte chloroplasts, despite the large differences in pigment composition. This finding largely rejects the hypothesized different evolutionary origin of plastids in freshwater raphidophytes. The chloroplasts of raphidophyceae were closely related to phaeophyceae agreeing with the phylogenetic position of these two groups based on nuclear genes [39]. 

The origin of unusual accessory pigments in *G. semen* and *V. virescens*, such as alloxanthin, diadinoxanthin, trans-neoxanthin and cis-neoxanthin, however, remains uncertain. Different pathways for the biosynthesis of alloxanthin, which is characterized by two acetylenic groups, have been suggested in previous studies. In cryptophytes, transformations of zeaxanthin or diatoxanthin into alloxanthin by hitherto unknown enzymes have been hypothesized [40,41,42]. In the dinoflagellate *Amphidinium carterae*, the use of a ^14^C-tracer has demonstrated that acetylene in alloxathin likely forms from an allen group [43]. Zeaxanthin is first changed to neoxanthin via violaxanthin. Neoxanthin, which has one allen group, is then converted into diadinoxanthin, which has one acetylenic group. The presence of violaxanthin, neoxanthin and diadinoxanthin in the *G. semen* and *V. virescens* suggest a pathway similar to that of *A. carterae* for the synthesis of alloxanthin. 

To ultimately identify the biosynthesis pathway of xanthopylls in freshwater raphidophytes, their nuclear genome needs to be sequenced, where the responsible enzymes are likely encoded. Previous studies have shown that only few genes encoding the chloroplast proteome are located in the chloroplast genome [31,44,45], while the majority has been transferred to the nucleus. The proteins encoded in the nucleus are synthesized as precursor proteins in the cytosol and are imported post-translationally into the chloroplast [46]. Nuclear genes appear to be more affected by horizontal gene transfer than chloroplast genomes [2] suggesting that the enzymes involved in the synthesis of the unusual xanthophylls in freshwater raphidophytes might originate from horizontal gene transfer.

Despite the high similarity in chloroplast genome composition of *G. semen*, *V. viresens* and *H. akashiwo*, one notable difference was observed in this study. The genes *ChlL, ChlN and ChlB*, encoding the light-independent protochlorophyllide oxidoreductase (LIPOR), were found in *V. virescens* but are absent in *H. akashiwo* chloroplast genomes. These genes have also been reported from the marine raphidophyte *C. subsalsa* [31]. However, the presence of LIPOR genes in *G. semen* remains unfortunately uncertain due to the incompleteness of the chloroplast genomes in this study. Protochlorophyllide oxidoreductases are essential for the transformation of the pigment protochlorophyllide to chlorophyllide during chlorophyll *a* synthesis. Two different enzymes, which are both present in cyanobacteria, can catalyze this step: the light-dependent (POR) and light-independent protochlorophyllide oxidoreductase (LIPOR). As the name suggests, LIPOR facilitates the synthesis of chlorophyll *a* in the dark [47]. After primary endosymbiosis, the POR genes were transferred to the nucleus, while the LIPOR genes remained in the chloroplast genome or were lost in several microalgal lineages [47]. In the phyla *chlorarachniophytes*, euglenoids and haptophytes, the loss of LIPOR genes may have occurred early during establishment of these lineages. In contrast, in chlorophytes, rhodophytes, *cryptophytes*, heterokonts, and *chromerids*, LIPOR genes were only lost by individual taxa [47] resulting in large heterogeneity regarding chlorophyll *a* synthesis in these groups.

The presence of LIPOR genes might be a beneficial adaptation to growth under low light conditions as commonly found in many northern European lakes rich in organic matter and humic substances. Furthermore, high iron concentrations in these lakes [48] might facilitate the synthesis of an iron-requiring protein such as LIPOR. In contrast, the synthesis of LIPOR might be disadvantageous for many marine taxa in iron-depleted regions in the ocean [31]. The loss of the LIPOR enzyme might have also occurred in several microalgal lineages due to its inefficiency under aerobic conditions. Similar to the nitrogenase enzyme, from which the LIPOR enzyme evolved, it is very sensitive to oxygen, allowing chlorophyll *a* synthesis only under anaerobic conditions [49]. Many lakes in northern Europe with high water color experience stratification and hypoxia in bottom waters during summer. Freshwater raphidophytes such as *G. semen* migrate at night several meters into the meta- and hypolimnion to access soluble reactive phosphorus gradually released from the sediment in anoxic conditions [50]. During this migration and presence in the anoxic hypolimion, these microalgae could utilize LIPOR for additional chlorophyll *a* synthesis.

## Figures and Tables

**Figure 1 genes-10-00245-f001:**
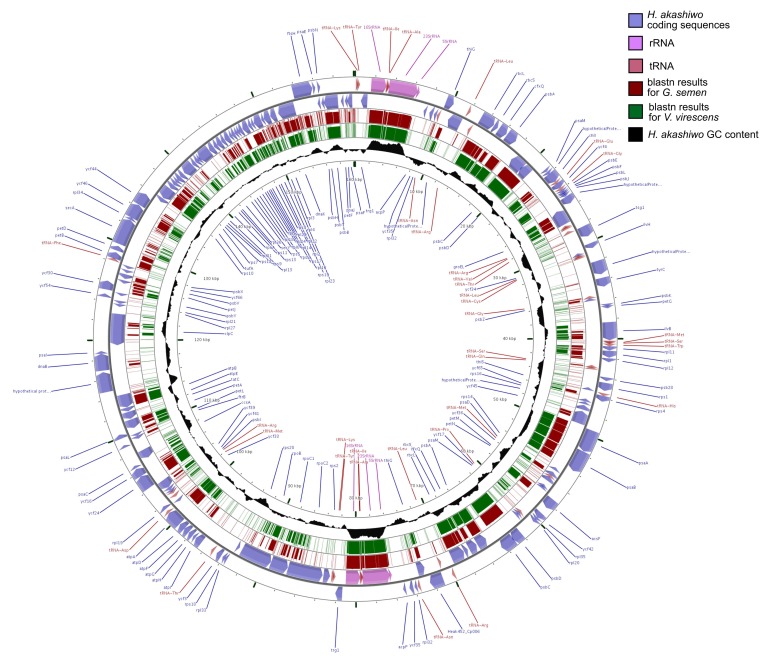
Sequence similarities based on blastn between to the circular chloroplast genome of *H. akashiwo* (~160 kbp) and chloroplast contigs of *G. semen* and *V. virescens* visualized with CGView. Coloured regions on the *G. semen* and *V. virescens* rings correspond to fragments matching the reference genome. The GC content of the *H. akashiwo* chloroplast genome averages at 30.5%.

**Figure 2 genes-10-00245-f002:**
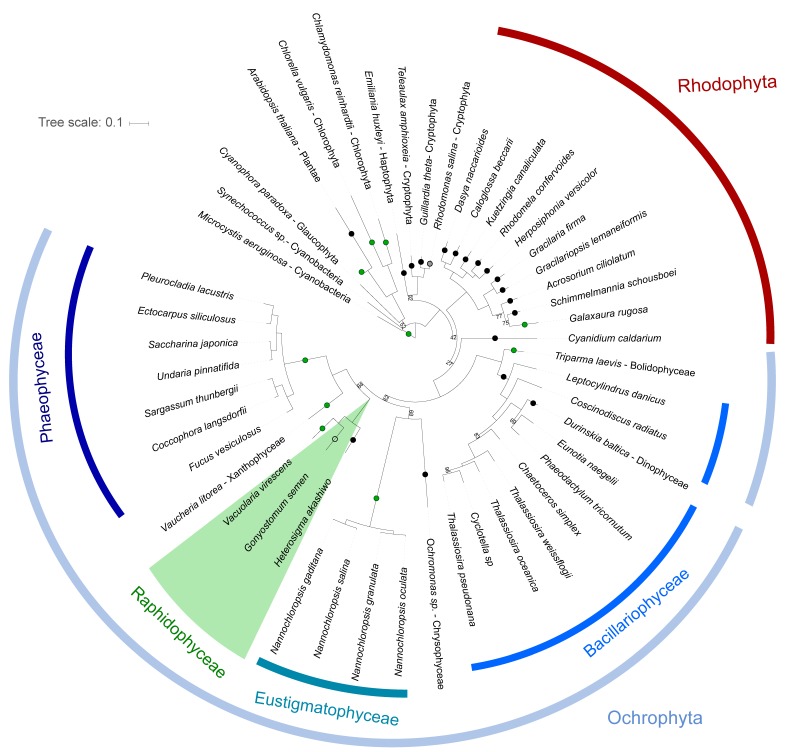
Maximum likelihood phylogeny of 53 concatenated chloroplast protein sequences, 14,896 amino acids in length, built with RAxML from an alignment of 49 taxa with bootstrap values <100 displayed on the internal branches. *Raphidophyceae* are highlighted in green, while curved lines indicate the position of other taxonomic groups. Circles on the branches indicate presence/absence of light-independent protochlorophyllide oxidoreductase (LIPOR) genes based on [31] and NCBI: green = presence, black = absence, grey = pseudogenes, empty = unknown.

**Table 1 genes-10-00245-t001:** Sequencing statistics of chloroplast (chl) samples from *G. semen* and *V. virescens*.

	*G. semen*	*V. virescens*
#reads (fwd and rev)	1,409,520	790,172
% bacterial reads	81.35	85.05
mean coverage of all contigs	95.3	75.4
#chl contigs	39	35
chl contigs length (bp)	74,603	88,279
mean coverage of chl contigs	612.9	472.3

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
