# Peer review of "Sequencing and Phylogenetic Analysis of Chloroplast Genes in Freshwater Raphidophytes"

_genes, 2019, doi:10.3390/genes10030245_

Round 1
Reviewer 1 Report
In the manuscript, the authors were using the whole genome amplification DNA from the algae species and trying to finish some chloroplast genes to conduct the phylogenetic analysis. it will supply some information about the kinds of method in chloroplast genome sequencing. But the results is not as good as what they addressed. For example, even they could minimize the effect from nuclear genome, but the over 80% of the reads from bacterial. It wasted lots of the data also. Based on the modern developed sequencing methods, the MDA is not the best choice. The author can do whole genome sequencing to get the full chloroplast genome also. Then they can get the nuclear genome data at the same time.
1.At the introduction parts, the authors should do some work to show all kinds of different sequencing methods used for whole chloroplast genome assembly. And why MAD is the best choice for them?
2.For the whole paper, please make a clear statement about chloroplast and plastid. Please do not use both at the same time. It is really confusing the readers.
3.In the method part, the authors describe the process for evaluate the chloroplast contigs, but they should make this part for better upstanding. Please make some pictures for how to choose the chloroplast read or contigs.
4.By using the MDA to get more chloroplast reads (also for bacterial reads). Why do not do more cultural and prepare more material instead? You can get more chloroplast DNA from this.
5.Please put more at the results for showing how to find the chloroplast contig.
6.In the discussion, please add more detail about this MDA vs other method for getting more chloroplast data.
Author Response
1.At the introduction parts, the authors should do some work to show all kinds of different sequencing methods used for whole chloroplast genome assembly. And why MAD is the best choice for them?
We regret that the reasoning for the chosen approach did not become clear throughout the manuscript. Multiple Displacement Amplification (MDA) is a very well established whole-genome-amplification method that we used to increase the amount of material for sequencing after organelle isolation. We decided to isolate the chloroplasts to exclude the raphidophyte nuclei, whose genome is likely several times larger than a human genome (see line 199), thus exceeding our financial resources for sequencing. We emphasized this reason again in line 56. Discussing advantages and disadvantages of different whole-genome-amplification methods is outside of the scope of this manuscript, but we mentioned limitations of MDA and alternatives in line 206-217.
The organelle genomes were sequenced on a standard Illumina MiSeq lane.
2.For the whole paper, please make a clear statement about chloroplast and plastid. Please do not use both at the same time. It is really confusing the readers.
We thank the reviewer for pointing out this confusion and clarified it in line 34-36. Like in most microalgal literature, we refer to photosynthetic plastids derived from secondary endosymbiosis as well as chloroplasts due to the common evolutionary origin and function.
3.In the method part, the authors describe the process for evaluate the chloroplast contigs, but they should make this part for better upstanding. Please make some pictures for how to choose the chloroplast read or contigs.
We thank the reviewer for pointing out this short-coming and expanded the paragraph about binning in line 82-85. Binning is a method commonly used in metagenomics to organize unconnected contigs into biologically relevant units such as draft genomes and plasmids thereby identifying near-complete novel genomes from environmental samples. There is extensive literature available about this approach and the steps included in the Anvio pipeline are detailed in the cited reference (Eren et al. 2015, PeerJ) and on http://merenlab.org/software/anvio/.
4.By using the MDA to get more chloroplast reads (also for bacterial reads). Why do not do more cultural and prepare more material instead? You can get more chloroplast DNA from this.
As mentioned under comment 1, we aimed at separating the chloroplasts from the nuclei to avoid sequencing the extremely large nuclear genome. At the beginning of this project, we also tried to employ density centrifugation to separate large amounts of chloroplasts from nuclei and cell debris. However, bursting the cells usually also damaged the chloroplasts making them unsuitable for density centrifugation. Thus, we didn’t find a way to separate large amounts of chloroplasts from nuclei requiring the utilization of whole-genome-amplification to reach the necessary amount of starting material for sequencing.
5.Please put more at the results for showing how to find the chloroplast contig.
Based on comment 3, we added more details to the paragraph about binning in Materials and Methods.
6.In the discussion, please add more detail about this MDA vs other method for getting more chloroplast data.
We thank the reviewer for this recommendation and added 2 more methods that might improve MDA for single cells in line 213-214. Furthermore, we mentioned already 2 other single cell whole-genome-amplification methods, which might be more suitable, in line 215.
Reviewer 2 Report
The paper presents new data and an interesting discussion about the evolution of plastid genomes in raphidophytes. It clearly has merit, but in my opinion could be improved in several ways. I would consider these minor revisions, though there are multiple of them.
Introduction:
lines 26, 27 - the wording in the first two sentences implies something that's simply not true:
"Chloroplasts are likely derived from a single endosymbiotic event incorporating a free-living cyanobacteria into the ancestor of all photosynthetic eukaryotic groups (glaucophytes, red algae and green algae)" - the term "chloroplasts" is used throughout the manuscript to describe not just green primary plastids, but others as well. Moreover, archaeplastida (comprising glaucophytes, red and green algae), are not ALL photosynthetic eukaryotic groups. Stramenopiles, dinoflagellates, crypto/haptophytes etc. all are unrelated to archaeplastida but are, too, photosynthetic eukaryotes, as the authors continue to state. The statements at the beginning of the introduction contradict each other and must be cleared up. Especially a non-phycological reader would be very confused by this.
minor typo: "unicellulare" instead of "unicellular" on line 36.
Methods:
All analyses should be reproducible according to methods. The alignment parameters need to be described in more detail. If defaults were used in Geneious, that's fine, but that should be stated.
The phylogenetic methods need to be described in more detail too - information on model of evolution implemented in RAxML, bootstrapping information (how many pseudoreplicates, was it the rapid bootstrap or regular?), etc.
Results:
the NCBI number should be added before publication.
Fig 1: it should be more clearly indicated how the incompleteness of the new genomes is shown in the figure. What marks the edges of separate contigs? If all the green and brown blocks and lines are separate contigs, then that would look like there are hundreds of them (rather than the reported 30-ish in table 1), and some of them don't correspond to coding sequences in the respective genomes. This needs to be clarified - maybe a zoomed-in inset in the figure to explain the different features? It is possible that I'm not reading the figure correctly, but that would just emphasize my point: the figure legend and caption, and the figure itself, needs to be presented more clearly and informatively.
Fig 2: caption also needs to be more informative
Discussion:
the discussion is very interesting, but I think the authors should make a better/clearer case for the absence of LIPOR genes in H. akashiwo - after all, the genome they recovered was fragmentary, so the absence of the genes in the data does not immediately imply absence in the genome. I am not sure from the methods section: it would be convincing to me if the reads were mapped to reference LIPOR sequences to see if there were any matches that did not assemble into de-novo contigs. If the authors did this (it is possible that I missed it), they should again emphasize it in the discussion to strengthen their argument.
Author Response
Introduction:
lines 26, 27 - the wording in the first two sentences implies something that's simply not true:
"Chloroplasts are likely derived from a single endosymbiotic event incorporating a free-living cyanobacteria into the ancestor of all photosynthetic eukaryotic groups (glaucophytes, red algae and green algae)" - the term "chloroplasts" is used throughout the manuscript to describe not just green primary plastids, but others as well. Moreover, archaeplastida (comprising glaucophytes, red and green algae), are not ALL photosynthetic eukaryotic groups. Stramenopiles, dinoflagellates, crypto/haptophytes etc. all are unrelated to archaeplastida but are, too, photosynthetic eukaryotes, as the authors continue to state. The statements at the beginning of the introduction contradict each other and must be cleared up. Especially a non-phycological reader would be very confused by this.
We thank the reviewer for pointing out this confusing paragraph. We clarified in line 27 that primary endosymbiosis resulted in the incorporation of plastids only in the archaeplastida and not in all photosynthetic eukaryotes. However, like in most microalgal literature, we refer to photosynthetic plastids derived from secondary endosymbiosis as well as chloroplasts due to the common evolutionary origin and function. We clarified our use of the term in line 34-36.
minor typo: "unicellulare" instead of "unicellular" on line 36.
We corrected this typo.
Methods:
All analyses should be reproducible according to methods. The alignment parameters need to be described in more detail. If defaults were used in Geneious, that's fine, but that should be stated.
We thank the reviewer for pointing this out and added the missing information in line 102.
The phylogenetic methods need to be described in more detail too - information on model of evolution implemented in RAxML, bootstrapping information (how many pseudoreplicates, was it the rapid bootstrap or regular?), etc.
We thank the reviewer for pointing this out and added the missing information in lines 109-111.
Results:
the NCBI number should be added before publication.
We’re currently submitting the sequences to NCBI and will add the accession numbers as soon as they are available.
Fig 1: it should be more clearly indicated how the incompleteness of the new genomes is shown in the figure. What marks the edges of separate contigs? If all the green and brown blocks and lines are separate contigs, then that would look like there are hundreds of them (rather than the reported 30-ish in table 1), and some of them don't correspond to coding sequences in the respective genomes. This needs to be clarified - maybe a zoomed-in inset in the figure to explain the different features? It is possible that I'm not reading the figure correctly, but that would just emphasize my point: the figure legend and caption, and the figure itself, needs to be presented more clearly and informatively.
We agree with the reviewer that the description of Figure 1 was insufficient. The dark red and green bars (G. semen and V. virescens) indicate sequence fragments that match the reference chloroplast genome of H. akashiwo based on blastn. The edges don’t mark thus the ends of the contigs, but the end of high sequence similarity. We added this explanation to the caption and legend of the figure. Matches in non-coding regions further support the high evolutionary relatedness of the freshwater and marine raphidophytes.
Fig 2: caption also needs to be more informative
We thank the reviewer for pointing out this short-coming and modified the caption of this figure. The caption reads now: “Maximum Likelihood phylogeny of 53 concatenated chloroplast protein sequences, 14,896 amino acids in length, built with RAxML from an alignment of 49 taxa with bootstrap values <100 displayed on the internal branches. Raphidophyceae are highlighted in green, while curved lines indicate the position of other taxonomic groups. Circles on the branches indicate presence/absence of LIPOR genes based on [31] and NCBI: green=presence, black=absence, grey=pseudogenes, empty=unknown.”
Discussion:
the discussion is very interesting, but I think the authors should make a better/clearer case for the absence of LIPOR genes in H. akashiwo - after all, the genome they recovered was fragmentary, so the absence of the genes in the data does not immediately imply absence in the genome. I am not sure from the methods section: it would be convincing to me if the reads were mapped to reference LIPOR sequences to see if there were any matches that did not assemble into de-novo contigs. If the authors did this (it is possible that I missed it), they should again emphasize it in the discussion to strengthen their argument.
We would like to clarify this issue. The H. akashiwo chloroplast genome was actually downloaded as a reference for the freshwater raphidophytes from NCBI and represents a complete circular genome. At least 2 different studies (Cattolico et al. 2008 BMC genomics, Seoane et al. 2017 Genome announcements) have sequenced entire chloroplast genomes from this species by now and all assemblies lack LIPOR genes.
We tried to clarify the presence or absence of LIPOR genes in G. semen by mapping the raw reads from both G. semen samples against the chloroplast contigs of V. virescens, but no reads mapped to the Vacuolaria contigs containing the LIPOR genes. This might be due to the absence of these genes or failed amplification, thus, not clarifying the issue. We added these information to the manuscript in lines 99-101.
Reviewer 3 Report
The paper describes plastid genome sequence of freshwater Raphidophytes obtained after whole genome amplification. Although the plastid genome is incomplete, it provided enough information to perform phylogenomic analysis and to identify some genes of the chlorophyl biosyntesis which are missing in marine relatives. The paper reads well, but still I have several coments and reccomendations. In general, it is not very common to publish incomplete plastid genomes, so the authors should show the effort they made to get as complete sequence as possible. Especially in the sequence assembly process and annotation.
1) The figure 1 lacks many details like the length of the complete H. akasiwo genome on which the new sequences were mapped (or at least a few genome coordinates), description of genes - it is common to show names of genes in plastid maps. Also GC content graph lack some explanation - does it apply to H. akasiwo or to new species? What is the maximum GC content?
2) It would be nice to provide a (supplementary) table containing the full list of genes identified in newly sequenced plastids and also the comparison with gene repertoire of marine relatives.
3) In the methods, I am missing details about SPAdes settings, RAxML settings (what subtitution matrix was used), Trimal etc. Different settings of programs can affect the results.
4) Did authors try some alternative approaches of plastid genome assembly? Maybe it would yield more complete genomes.
5) The phylogenetic tree is rather difficult to read, I suggest to display rooted tree and only branch supports below 100. Moreover, the assignment into higher taxonomical groups is not displayed uniformly. It would be also nice to depict in the picture the losses of LIPOR genes as these genes are discussed in the text.
6) There is probably a typo in the text (line 158) - the phylogenetic matrix contained 14,896 amino acids, not 14,896bp.
7) Did authors try to search for LIPOR genes in transcriptomic data available for Chatonella marina and Heterosigma akasiwo to detect potential horizontal gene transfer?
8) Did authors try to search for LIPOR genes in G. semen by another method? For example by PCR with primers designed based on V. virescens plastid sequence.
Author Response
1) The figure 1 lacks many details like the length of the complete H. akasiwo genome on which the new sequences were mapped (or at least a few genome coordinates), description of genes - it is common to show names of genes in plastid maps. Also GC content graph lack some explanation - does it apply to H. akasiwo or to new species? What is the maximum GC content?
We thank the reviewer for pointing out this short-coming, adjusted the caption and legend of figure 1, and indicated the location of genes. We initially didn’t include the genes in the figure, as we worried about making the figure too crowded. The caption reads now: ”Sequence similarities based on blastn between to the circular chloroplast genome of H. akashiwo (~160kbp) and chloroplast contigs of G. semen and V. virescens visualized with CGView. Coloured regions on the G. semen and V. virescens rings correspond to fragments matching the reference genome. The GC content of the H. akashiwo chloroplast genome averages at 30.5%.”
2) It would be nice to provide a (supplementary) table containing the full list of genes identified in newly sequenced plastids and also the comparison with gene repertoire of marine relatives.
We thank the reviewer for this recommendation and provide such a table as supplementary information.
3) In the methods, I am missing details about SPAdes settings, RAxML settings (what subtitution matrix was used), Trimal etc. Different settings of programs can affect the results.
We regret that we didn’t include this information initially and added them to the new version of the manuscript. Assembly settings are specified now in lines 77-79. Additional information about settings in trimAl can be found in line 103. The RAxML settings are detailed in line 109-111.
4) Did authors try some alternative approaches of plastid genome assembly? Maybe it would yield more complete genomes.
As mentioned in line 76, 80-81 and 87-91, we used IDBA-ud, SPAdes and metaSPAdes, as well as reduced datasets and homogenized read depth to improve the assemblies. These different approaches were unfortunately not very successful and improved the assemblies only marginally.
5) The phylogenetic tree is rather difficult to read, I suggest to display rooted tree and only branch supports below 100. Moreover, the assignment into higher taxonomical groups is not displayed uniformly. It would be also nice to depict in the picture the losses of LIPOR genes as these genes are discussed in the text.
We thank the reviewer for this recommendation and improved the tree accordingly. The tree is currently rooted with the 2 cyanobacterial species. We changed the display of taxonomic groups to the phyla Rhodopyta and Ochrophyta, as well as the major classes within Ochrophyta. We also indicated the presence and absence of LIPOR genes in the represented taxa.
6) There is probably a typo in the text (line 158) - the phylogenetic matrix contained 14,896 amino acids, not 14,896bp.
Yes, we corrected this typo.
7) Did authors try to search for LIPOR genes in transcriptomic data available for Chatonella marina and Heterosigma akasiwo to detect potential horizontal gene transfer?
2 previous studies have sequenced complete chloroplast genomes of Heterosigma akashiwo (Cattolico et al. 2008 BMC genomics, Seoane et al. 2017 Genome announcements) and did not detect any LIPOR genes, thus reliably indicating the lack of these genes in this marine raphidophte species. In contrast, unpublished data in Hunsperger et al. 2015 (BMC Evolutionary Biology) suggest presence of these genes in the chloroplast genome of Chattonella subsalsa.
8) Did authors try to search for LIPOR genes in G. semen by another method? For example by PCR with primers designed based on V. virescens plastid sequence.
We tried to clarify the presence or absence of LIPOR genes in G. semen by mapping the raw reads from both G. semen samples against the chloroplast contigs of V. virescens, but no reads mapped to the Vacuolaria contigs containing the LIPOR genes. This might be due to the absence of these genes or failed amplification, thus, not solving the issue. We added these information to the manuscript in lines 99-101.
Designing LIPOR specific primers and applying these to G. semen samples would definitely be a promising approach to clarify the presence or absence of the genes. Unfortunately, the lead-author of this study is currently working at a different university and it is, thus, difficult to conduct any further laboratory work for this project.
Round 2
Reviewer 3 Report
The manuscript has been edited according to the comments, however, I think the Figure 1 is still rather difficult to read. Maybe it is just in the PDF version I obtained for the review, but if not, then authors should find another way how to present the plastid genome maps.
All other points were addressed.
Author Response
We thank the reviewer for pointing out the low quality of Figure 1. We improved it and uploaded a new version.